# GCINT: Dynamic Quantization Algorithm for Training Graph Convolution Neural Networks Using Only Integers

## Abstract

Quantization approaches can minimize storage costs while decreasing the computational complexity of a model, although there is minimal study in the GNN field on quantization networks. We studied the four primary reasons why existing quantization approaches cannot be employed extensively with GNNs: (1)Quantifying the distinctions between data sources; (2)Quantifying the distinctions between data streams; (3)Quantifying the distinctions between concentrations; (4)QAT's Limitations. Based on this, we propose GCINT, which is an efficient quantization framework prepared for GNN training. The entire forward, backward, optimizer, and loss functions are calculated using integer data. We achieved a training acceleration ratio of nearly 10× compared to FP32 Cuda Core in RTX 2080TI INT8 Tensor Core. Our quantization is independent of the dataset and weight distribution, and more than 2,000 randomized trials have been undertaken on the 8 popular GNN benchmark datasets, with all achieving errors within 1% of the FP32.

## 1 INTRODUCTION

There is an abundance of graph-structured data in the natural and social sciences. In fields such as social networks (Fan et al., 2019), recommender systems (Wu et al., 2020), traffic networks (Jiang & Luo, 2022), molecular prediction (Mansimov et al., 2019), and drug discovery (Zhang et al., 2022), Graph Neural Networks (GNNs) representative deep learning systems for graph data learning, inference, and generalization have produced superior outcomes. As graph learning applications increase and graph data expands, the training of GNNs becomes inefficient due to two significant obstacles: (1) **Storage expenses.** Since training needs recording the outputs of several layers in forward propagation for backward propagation calculation, extremely large scale graph data is frequently saved utilizing distributed CPU-centric memory storage by distributed GPU clusters employing a mini-batch training technique. Common acceleration devices such as GPUs and FPGAs with on-chip storage and bandwidth can no longer match the demand for training large GNNs and are too dependent on sampling techniques to train on a device with a limited batch size for each training session (Yang, 2019). (2) **Calculated expenses.** Training a single epoch on the Reddit dataset generally requires tens of TFLOPS, even for KB-sized GNN models.

Quantization (Yang et al., 2019) can lower storage costs while decreasing the model's computational complexity (Nagel et al., 2021). Although quantization is widely used in CNNs, research on quantized networks for GNNs is scarce, we believe the following factors primarily restrict the applicability of quantization approaches in GNNs: (1) **Quantifying the distinctions between data sources.** During CNN training, the RGB images of UINT8 are normalized and sent to the network. In contrast, when using GNN models, the node features of the network are frequently not the consequence of normalization, and the distribution of node features will shift as the graph changes and embedding methods are employed. The information contained in the image could have been represented by UINT8, whereas the embedding vectors of graph nodes are typically in FP32 data format, which contains significantly more information than UINT8. Therefore, it is essential for GNN to quantize the dataset, which must represent a large amount of information in the dataset with a limited number of bits. (2) **Quantifying the distinctions between data streams.** The calculation in each layer of CNN that maps to the GPU is typically General Matrix Multiplication (GEMM), and the activation distribution in each layer of GNN is strongly tied to the graph topology, when the

average degree of the graph is high, the aggregation process of the integer domain is more susceptible to data overflow. Conversely, when the average degree of the graph is low, the distribution of activation in each layer of the network will be focused in the low bit data range. This brings uncertainty into quantitative training, and typical quantization approaches for CNN will not be employed directly in GNN models. (3) **Quantifying the distinctions between concentrations.** Due to the fact that DNN models generally include millions of parameters, it is required to decrease the complexity of storage and processing by quantizing and compressing the weights, such as BNN (Tang et al., 2017) and XNOR-Net (Rastegari et al., 2016) quantize the weights to binary. However, GNN models are typically in the KB order of magnitude, and the gains from compressing the weights are not substantial. (4) **QAT's Limitations.** The majority of CNN quantization is based on the research of QAT (Wang et al., 2019) quantization operators, and this design strategy has evolved into a robust, low-error quantization model. QAT conducts low-bit quantization of weights and activation during forward propagation, then reduces the noise and loss induced by forward quantization using FP32 back propagation, and lastly dequantizes the model to integer for inference acceleration during model deployment (Krishnamoorthi, 2018). The restriction of the QAT is that it cannot be utilized for accelerated training since the data format after quantization is still FP32 during the training process. After training a CNN model for a certain class of tasks with a significant quantity of data, they seldom need to be retrained after deployment, hence QAT provides CNN models with extraordinarily high advantages. GNNs tend to be dynamic graphs in the real world and require fine-tuning or retraining of the model; hence, speeding the training process of GNNs is more relevant than accelerating the inference process, which QAT cannot perform for accelerated training.

This work considers the motivations and problems associated with quantization of graph architectures, and provides the following contributions:

- We employ a top-down quantification study methodology. The vast majority of prior quantization investigations have been bottom-up procedures, i.e., beginning with the FP32 tensor, quantizing to the FP32 tensor that can be mapped by integers one by one, and then dequantizing to the integer, which we consider a superfluous operation. Consequently, we investigate the quantization strategy for integer tensor computation directly from a different angle, which has the advantage that a shaping model can be obtained directly without dequantization and can be used directly for inference and training speedup in fixed-point hardware, such as GPU INT Tensor Core.

- We propose a novel quantization training algorithm for graphs as an alternative to the traditional QAT method to accelerate the GNN training process. This algorithm can adaptively adjust the quantization range according to the sparsity of the graph data and can accommodate unevenly distributed data during training. The entire training forward, backward, optimizer, and loss functions are calculated using integer data, which can be directly accelerated by using the INT Tensor Core of GPU for GNN training. We achieved a training acceleration ratio of nearly $10\times$ compared to FP32 Cuda Core in RTX 2080TI INT8 Tensor Core, and can train a larger subgraph than the original one with limited memory.

- Our quantization is independent of the dataset and weight distribution, and more than 2,000 randomized trials have been undertaken on the 8 popular GNN benchmark datasets, with all achieving errors within $1\%$ of the FP32 baseline, and without fine-tuning hyperparameters.

## 2 BACKGROUND AND RELATED WORK

Table 1 presents the quantitative studies published in recent years. **WAGE** (Wu et al., 2018) provides a low-bit quantization approach for weight, activation, gradient, and error in CNN training. Data is linearly mapped using integers to FP32, and all training data must be dequantized to the integer domain before being utilized for training acceleration. The authors believe that following matrix multiplication of the $bita$ tensor with the $bitw$ tensor, an $bit[a + w - 1]$ tensor is formed, which is then quantized to the $bita$ tensor and sent to the next network layer. We feel that basing the quantization approach on this assumption is not rigorous since it disregards the dimensionality of the tensor. Assuming that the dimensionality of the tensor is $n$, the data bit-width range of the output tensor should be $[a+w-1+log_2n]$. However, because the GEMM and SPMM(Sparse-Dense Matrix Multiplication) dimensions in GNN are typically large, the output data bit-width range will overflow to varying degrees, rendering the method inapplicable to the training speedup of GNN.

Table 1: Comparing the quantization precision of several methods. **PUB** indicates the origin of the algorithm and the date of its publication. Following is the number of **W**(Weight), **A**(Activation), **G**(Gradient), **E**(Error) and **OPTIM**(Optimizer) bits after the respective quantization. **SFB**(Save For Backward) is the number of bits of the forward that are saved to compute the back propagation during training. Where $32(n)$ represents a floating-point number quantized to $bit n$(stored and calculated in FP32 format, the number will be dequantized to yield INT $n$ integers). **TRAIN** shows if training may be expedited (forward and backward). **INFER** specifies whether or not inference may be accelerated. **DEQ**(Dequantization, FP to INT) specifies whether dequantization is required by the algorithm. **DOMAIN** specifies the algorithm's application space and the scope of the experiment.

| METHOD | PUB | W | A | G | E | SFB | LOSS | OPTIM | DEQ | TRAIN | INFER | DOMAIN |
|--------|-----|---|---|---|---|-----|------|-------|-----|-------|-------|--------|
| **WAGE** | ICLR2018 | 32(2) | 32(8) | 32(8) | 32(8) | 32(8) | FP32 | FP32 | ✔ | ✔ | ✔ | CNN |
| **DOREFA** | CORR2018 | 32(1) | 32(4) | FP32 | FP32 | FP32 | FP32 | FP32 | ✔ | ✘ | ✔ | CNN |
| **DFP** | ICLR2018 | 16 | 16 | 16 | 16 | 16 | FP32 | FP32 | ✔ | ✔ | ✔ | CNN |
| **DQ** | ICLR2021 | 32(8) | 32(8) | FP32 | FP32 | FP32 | FP32 | FP32 | ✔ | ✘ | ✔ | GNN |
| **EXACT** | ICLR2022 | FP32 | FP32 | FP32 | FP32 | 2 | FP32 | FP32 | ✔ | ✘ | ✘ | GNN |
| **GCINT** | THIS WORK | 8 | 8 | 8 | 8 | 8 | 8 | 8 | ✘ | ✔ | ✔ | GNN |

**DoReFa-Net** (Zhou et al., 2016) proposes a lower-bit weight, activation quantization scheme, which introduces a nonlinear activation function in the computation to improve the state representation of the network for overflow data. However, the process is still presented in FP32, and dequantization becomes more complex due to nonlinear activation. In general, it is difficult to directly implement DoReFa-Net training on integer-based hardware, although it has the potential to be utilized in the exploration of high-dimensional discrete spaces with discrete gradient descent directions.

**DFP** (Das et al., 2018) proposes the method of employing mixed precision training with integers in CNNs so that precision-critical operations (such as the optimizer, normalization, and loss function) are maintained at FP32 while computationally intensive processes are maintained at INT16. Since the result of the multiplier output of INT16 will create up to 30 bits of data with 1 bit of sign, at the time of MAC, at least the accumulation unit of INT32 is necessary to prevent data overflow. The computation result is transformed into an FP32 tensor output in the DFP. Before the next layer of input, the FP32 tensor must be dequantized into an INT16 tensor. This ensures that the accumulation of data will not result in a massive excess of integers. To participate in the subsequent phase of training, the output of the loss calculation and optimizer parameter update phase must be dequantized from the FP32 tensor to the INT16 data format. DFP investigates the concept of integer training, which enables intense computing to be directly utilized in integer-based hardware. The training is still comprised of a substantial number of FP32 dequantization and is not an entirely integer training stream.

**DQ** (Tailor et al., 2020) proposes QAT method for GNN. The authors believe that the largest source of quantization error is the portion of the node with a higher degree, because the node degree in the graph exhibits a power-law distribution, resulting in a larger output in the aggregation of the portion of the node with a higher degree, which necessitates a larger bit width, leading to a greater variance of the entire batch of data and a very large error when the bit width is converted downward. DQ has effectively analyzed the issues that occur from quantizing GNN, assigning the high-degree node to the FP32 Tensor Core and the low-degree node's data to the INT Tensor Core, and the task assignment of hardware resources will vary for different graph as memory management changes with the degree, makes it challenging to deploy the actual solution to the GPU side for inference acceleration. On the other hand, large graphs are frequently trained with a fixed node sampling method for MiniBatch, in which case there is no variation in the degree of each node, rendering DQ inapplicable; hence, DQ is incompatible with existing graph sampling algorithms.

**EXACT** (Liu et al., 2021) provides a low-memory training approach for GNNs, and the author's concept is truly groundbreaking. EXACT utilize FP32 format for both forward and backward propagation; however, a portion of the backward propagation computation must utilize some of the activation saved in the forward pass. The authors quantize the data to a very low level of precision and store it in the memory before dequantization to FP32 in order to perform the operation with the

error tensor in FP32 format. This drastically reduces the amount of space required for training, but it cannot be utilized to speed up training.

We propose **GCINT**, a quantization approach that accelerates graph training and reduces memory consumption. It may be utilized directly in the GPU's INT Tensor Core to accelerate training without the need for dequantization procedures.

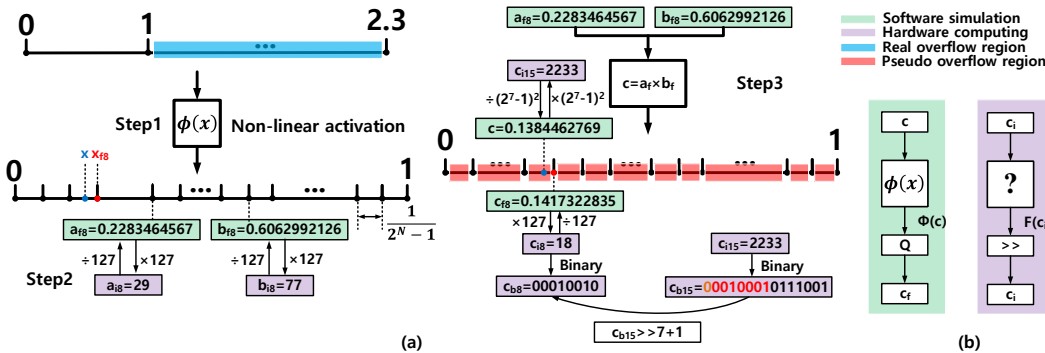

Figure 1: (a) and (b) depict the fundamental quantization and dequantization procedures. The green region represents the floating-point data stream for simulated quantization, the purple region represents the integer data stream for actual fixed-point hardware execution, the blue region represents the real overflow, and the red region represents the pseudo overflow.

## 3 METHODOLOGY

### 3.1 THE IGNORED PSEUDO-OVERFLOW ISSUE

Before proposing the solution, we first generate the pseudo-overflow problem, which is not accounted for in conventional quantization algorithms, and discuss how nonlinear activation influences dequantization. We use $x$ to represent any arbitrary scalar that is compatible with the FP32 data format. $x_{fn}$ represents the number of signed $bitn$ to which $x$ is quantized (as a floating-point value), $x_{in}$ represents the integer representation of $bitn$, and $x_{bn}$ represents the binary representation of $bitn$. From Figure 1(a) $a_{f8}$, although it is a floating-point representation, that data falls squarely on the data endpoint that divides the 0-1 interval into 127 copies, indicating that $x_{in}$ multiplied by $2^n - 1$ must be an integer. Figure 1(a) depicts the typical scheme's quantization and dequantization data flows. The pre-computation quantization and post-computation quantization of a batch of data typically need three phases, and INT8 quantization serves as an illustration. A nonlinear activation function will first remap the data to the interval [-1,1], and the second step quantifies the $x$ in the interval to its nearest endpoint split by area 127, i.e., the $x$ to $x_{f8}$. In the third phase, all $x_{f8}$ are utilized to do the necessary computation. If the computation involves vectors or matrices, the outcome may be more than 1. So, following each calculation, step1 will be assigned a nonlinear mapping to the interval [-1,1], and then re-quantized.

In conventional quantization, the $x_{fn}$ (green region) is often employed for training or inference. To really accomplish computational acceleration with fixed-point hardware, it is frequently essential to dequantize the data in the green area to the integer domain in order to generate a data stream consisting of just integers (the purple region is the data executed in the fixed-point hardware). In Figure 1(a), step3 we require the following stage of input as $c_{f8}$. Quantizing $c$ to $c_{f8}$ corresponds to shifting the binary field by 7 bits and then plus 1 ($c_{b15}$ to $c_{b8}$), regardless of the bit-width of the accumulator, they will round off the lowest 7 bits of the result and keep the next lowest 7 bits to form a new data (red portion of $c_{b15}$), plus a sign bit (orange portion of $c_{b15}$) to form a new INT8 integer.

**Real overflow:** refers to the calculated data portions that exceed 1 (blue area in step1). Existing quantization often resolves the problem of genuine overflow using a variety of nonlinear activation functions, such as $clamp(X, -1, 1)$ in the Google White Paper (Krishnamoorthi, 2018), $tanh$ in

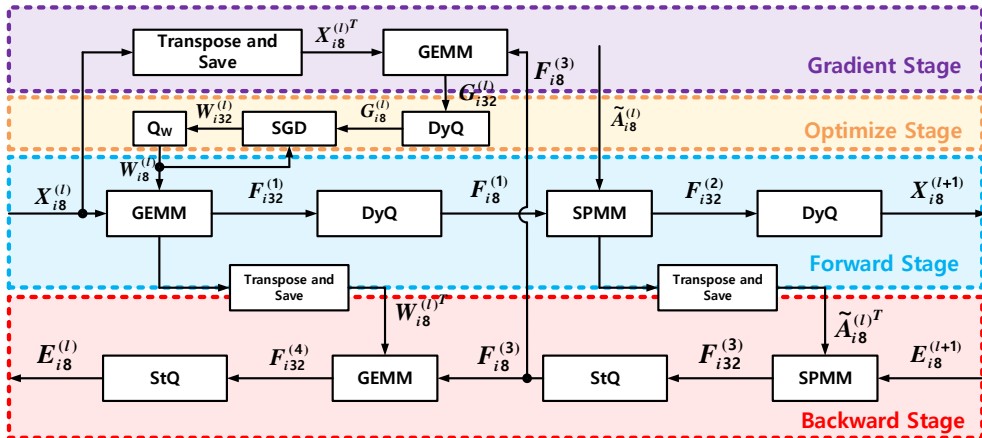

Figure 2: GCINT's network layer. For the sake of clarity, the ReLU and Dropout layers are omitted from the figure. $X_{in}^{(l)}$, $W_{in}^{(l)}$, $E_{in}^{(l)}$, $G_{in}^{(l)}$, $\tilde{A}_{in}^{(l)}$, $F_{in}^{(k)}$ are tensors in which features, weights, errors, gradients, and normalized adjacency matrices are represented as INT $n$ signed integers. $F_{in}^{(k)}$ denotes the tensor representation inside the $l^{th}$ layer. At the moment of initialization, the integerized embedding tensor $X_{in}^{(0)}$ and $\tilde{A}_{in}^{(l)}$ (dataset) are kept in memory. The mode of initiation: $X_{in}^{(0)} = \lceil \frac{X^{(0)}}{max(|X^{(0)}|)} \times (2^n - 1) \rceil$, $\tilde{A}_{in}^{(l)} = \lceil \tilde{A}^{(l)} \times (2^n - 1) \rceil$.

DoreFa-Net (Zhou et al., 2016) , etc. The primary issues are as follows: (1) **Universal.** Quantization is optimal when the majority of the data distribution is concentrated in the area with the highest activation function derivative, as there are more state representations at this point. No totally universal nonlinear activation function exists to handle the actual overflow problem for diverse data sets or even for data streams from different network tiers. (2) **Whether the dequantization function has a solution is an open question.** After computing $c$ in step3, as seen in Figure 1(b), the algorithm will employ nonlinear activation to remap the data to $\phi(c)$ and then quantize it. Since the hardware output is $c_{in}$, the dequantization function $F(x)$ of $\phi(x)$ must satisfy the following conditions, $\phi(c)$ is equal to the quotient of $F(c)$ and another integer, and the computation in $F(x)$ cannot contain a floating-point operation; otherwise, the one-to-one mapping between $c_{fn}$ and $c_{in}$ is not satisfied after the quantization or shift operation. Therefore, when the quantization interval is nonlinear, it is frequently challenging to directly dequantize to an integer, and low-bit computational speedup cannot be achieved because the dequantization of $\phi(x)$ does not necessarily have a solution to prove the existence or nonexistence of $F(x)$.

**Pseudo overflow:** Here, we define pseudo-overflow as the occurrence in which the result of multiplying and adding the $x_{fn}$ cannot be stated using $2^n - 1$ (Figure 1(a) red area). As for $a_{f8}$ and $b_{f8}$, the result $c$ does not fall on the endpoints divided into 127 copies with a 0-1 interval, indicating that a finer division of the 0-1 interval (more data blocks larger than 127) is required to express $c$ directly, such that $c$ needs to divide 0-1 into 16129 ($2^{13} - 1 < 16129 < 2^{14} - 1$) parts, it requires at least 14 bits. Therefore, despite the fact that the data do not exceed the 0-1 interval, a severe overflow has occurred. Pseudo-overflow in the floating-point is not as intuitive as actual overflow and is frequently overlooked by algorithm researchers. However, since a substantial quantity of data is frequently concentrated in the 0-1 calculation interval, pseudo-overflow can introduce extremely large errors to quantization.

### 3.2 INT8 DYNAMIC AND STATIC QUANTIZATION OPERATORS

Our works is based on the basic GCN (Kipf & Welling, 2016) layer, $X^{(l)}$ is the node embedding matrix at the $l^{th}$ layer, $W^{(l)}$ is the weight matrix of the $l^{th}$ layer, $\tilde{A} = \tilde{D}^{-\frac{1}{2}} A \tilde{D}^{-\frac{1}{2}}$ is the normalized adjacency matrix, where $\tilde{D}$ is the degree matrix of $A + I$, Since $\tilde{A}$ is often sparse and feature $X^{(l)}$

is typically dense, the GCN layer is expressed as the expression of GEMM and SPMM (Fey & Lenssen, 2019) , see Equation 1 .

$$X^{(l+1)} = ReLU \left( SPMM \left( \tilde{A}^{(l)}, GEMM \left( X^{(l)}, W^{(l)} \right) \right) \right) \tag{1}$$

We define the dynamic quantization (**DyQ**), static quantization (**StQ**), and weight quantization ($Q_w$) operators. GCINT's network layer is shown in Figure 2.The distinction between **DyQ** and **StQ** is that **DyQ** modifies the quantization interval based on the distribution of matrix, whereas **StQ** disregards. An all-integer network exhibits a pseudo-overflow phenomena, according to our findings. Multiplication of two $x_{f8}$ integers requires at least 15 bits (14 data and 1 sign) to completely describe the result without degradation, the result is deemed to have pseudo overflow. From Figure 3, clearly, when matrix multiplication is conducted using integers, the results(absolute value) in the region $[2^7, 2^{13}]$ correspond to the floating point pseudo overflow, whilst the real overflow occurs in the region beyond $2^{14}$, which corresponds to the portion of the data where the floating point is beyond than 1.

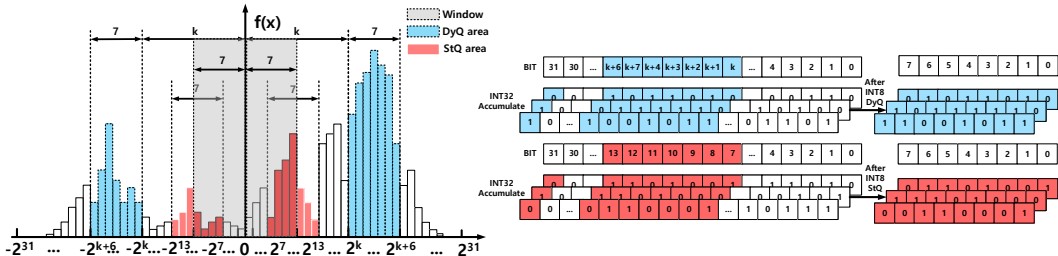

Figure 3: histogram of the output tensor of the matrix multiplication of two INT8s, where $f(2^x)$ represents the number of data in $[2^x, 2^{x+1})$. The process of converting INT32 accumulation to INT8 under **DyQ** and **StQ** is seen on the right one.

We utilize Kullback-Leibler divergence (Joyce, 2011) to prove that when $g(k)$ in Equation 2 gets the maximum, we may derive a 7-bit($[k, k+6]$) expression for this data set with reduced Kullback-Leibler divergence and more Shannon Entropy (Bromiley et al., 2004). From Figure 3, when quantifying to INT8, two windows of length 7 are required. Each time, they take one step in the other way. Before each step, the region encompassed by the window and the current step value are recorded. When the slide traverses each of the 25 steps, the step with the biggest area is selected as the $k$. Large area equals more data, therefore the 7 bits from the $[k, k+6]$ may effectively represent the majority of the matrix's data and can harvest more expressive status and distinction. If the conventional direct quantification method outlined in Figure 1, by default, the corresponding integers use the 7th to 13th bits as the expression of the whole batch of data. During the training process of GNN, the sparsity of the graph and the initialization of the weights are different, causing GEMM and SPMM to be heavily distributed between the real overflow region ($> 2^{14}$) and the pseudo overflow region ($< 2^{14}$), or between the real and pseudo, and if a fixed truncation pattern is adopted ( $> 2^{14}$ takes 127, $< 2^7$ takes 0), resulting in a poor quantization effect when the data is unevenly distributed.

$$g(k) = \sum_{i=0}^{6} f \left( 2^{k+i} \right) + f \left( -2^{k+i} \right) k = 0, 1, ..., 25 \tag{2}$$

In Equation 3 and Equation 4, we display both the binary and decimal representations of the dynamic quantization function. Since computers utilize binary storage and processing, our GPU implementation utilizes Equation 3's operations mostly in the form of bitwise operations. Where 7'b represents 7-bit binary and "|" represents the bit-by-bit "OR" operation on consecutive binary. The StQ function is equivalent to the case where $k$ equals 7 in **DyQ**. Algorithm 1's code demonstrates the GPU computation of $k$, whose time complexity is significantly less than that of matrix multiplication ($O(logn) \ll O(n^3)$), and whose high parallelism facilitates parallel processing in the GPU.

$$Binary : DyQ(X_{b32}) = \begin{cases} concat(X_{b32}[31], 7'b1111111) & |X_{b32}[30:k+7] = 1 \\ concat(X_{b32}[31], X_{b32}[k+6:k]) & otherwise \end{cases}$$

(3)

$$Decimal : DyQ(X_{i32}) = \lceil \frac{clamp\left(X_{i32}, -2^{k+7}, 2^{k+7}\right)}{2^k} \rceil$$

(4)

---

**Algorithm 1:** Calculate Parameter $k$ of **DyQ** on GPU. Maxbit($X_{b32}$): Remove the sign bit and return the location of the first occurrence of 1 from the high bit to the low bit for each element. ThreadGroup[32,$n^2$]: Allocate $32 \times n^2$ threads on the GPU.

---

**Input:** $X_{b32} \in R^{n \times n}$
**Output:** $k$

1 **InitializeMatrix** $\left( A_{b5} \in R^{1 \times n^2}, B_{b32} \in R^{1 \times 31}, C_{b32} \in R^{1 \times 26} \right)$
2 $A \leftarrow$ **Maxbit**($X_{b32}$).**flatten**() then **LoadAtoeachThreadGroup**()
3 **for** $i$=0 to 31 each ThreadGroup[i] **Parallel do**
4     **if** ThreadGroup[i][j]=i **then** ThreadGroup[i][j]$\leftarrow$ 1 **else** ThreadGroup[i][j]$\leftarrow$ 0
5     B[i]$\leftarrow$ **AdderTree**(HEIGHT=$2log_2n$)
6 **end**
7 **for** $q$=0 to 25 **Parallel do**
8     **for** $q$=0 to 6 **do**
9         | C[q]$\leftarrow$ C[q] +B[v + q]
10     **end**
11 **end**
12 $k \leftarrow$**Argmax**(C)
13 return $k$

---

### 3.3 QUANTIZATION OF ACTIVATIONS

In forward propagation, each matrix multiplication is followed by a dynamic quantization using Equation 5's computing process. In contrast to back propagation, the output tensor of each layer in the forward direction must ensure a high degree of discrimination and Shannon Entropy, and the data quantization interval should fall as close as possible to a position with a higher derivative of the quantization function so that the data can acquire more state bits. Stability is the greatest challenge of quantitative training, which is mostly caused by the disparity between graph distributions and weight initialization. **DyQ** can resolve this issue by using the findings of GEMM and SPMM to alter the quantization interval. As seen in Table 2, we initialize the matrices to be quantized by INT8 of the two quantization techniques with different distributions and determine the Shannon Entropy of the output matrix. We can see that **DyQ** handles uneven distributions effectively, achieves greater Shannon Entropy than classical(Figure 1) direct quantization for nearly all distributions, and achieves nearly the maximum amount of information that can be conveyed by 8-bit state bits under uniform distribution.

$$Forward : X_{i8}^{(l+1)} = DyQ \left( SPMM \left( \tilde{A}_{i8}^{(l)}, DyQ \left( GEMM \left( X_{i8}^{(l)}, W_{i8}^{(l)} \right) \right) \right) \right)$$

(5)

### 3.4 QUANTIZATION OF WEIGHTS

To assure the integerization of the complete stage, the computational portion of the optimizer must be as basic as feasible, and it must not contain any additional nonlinear functions; hence, we choose SGD (Bottou, 2012) as our optimizer(Equation 6). We must employ integer weights and a gradient tensor throughout the calculation of the optimizer. We continue to compute the difference between the weights and the gradient using the cumulative unit of INT32. After obtaining an updated weight, we normalize a new set of weights using an unsaturated quantization approach(Equation 7).

$$Optimize : \quad W_{i8}^{(l)} = Q_w \left( W_{i8}^{(l)} - \mu \times G_{i8}^{(l)} \right)$$

(6)

Table 2: N(A,B) denotes a normal distribution with mean A and standard deviation B, U(±T) denotes a uniform distribution in the interval [-T,T], and H(x) is the Shannon Entropy where $H(X) = -\sum_{i=1}^{n} p(x_i) \log p(x_i)$.

| Matrix Distrubution | $N(0, 2^{14})$ | $N(0, 2^7)$ | $N(2^5, 2^7)$ | $N(2^7, 2^7)$ | $N(2^{12}, 2^{14})$ | $N(2^{13}, 2^{14})$ | $N(2^{14}, 2^{14})$ | $N(2^{15}, 2^{14})$ | $N(2^{16}, 2^{14})$ |
|---|---|---|---|---|---|---|---|---|---|
| $H(DyQ(X))$ | **6.993** | **7.007** | **6.991** | **6.992** | **7.046** | **7.053** | **7.033** | **6.053** | **5.048** |
| $H(Trad(X))$ | 6.647 | 2.114 | 2.098 | 2.106 | 6.525 | 6.162 | 4.289 | 1.826 | 0.023 |

| Matrix Distrubution | $U(\pm 2^7)$ | $U(\pm 2^8)$ | $U(\pm 2^{10})$ | $U(\pm 2^{12})$ | $U(\pm 2^{14})$ | $U(\pm 2^{16})$ | $U(\pm 2^{20})$ | $U(\pm 2^{25})$ | $U(\pm 2^{31})$ |
|---|---|---|---|---|---|---|---|---|---|
| $H(DyQ(X))$ | **7.997** | **7.813** | **7.990** | **8.000** | **8.000** | **8.000** | **8.000** | **8.000** | **8.000** |
| $H(Trad(X))$ | 1.498 | 2.252 | 4.064 | 6.015 | 8.000 | 3.564 | 1.228 | 1.008 | 1.004 |

$$Q_w\left(W_{i32}^{(l)}\right) = \lceil \frac{W_{i32}^{(l)} \times 127}{\max \left|W_{i32}^{(l)}\right|} \rceil \tag{7}$$

## 3.5 QUANTIZATION OF ERROR

In contrast to forward propagation, all errors are quantized statically, as seen by the Equation 8. The error often does not require as much differentiation as the activation since the error tensor represents the direction and magnitude of each layer's offset. Even with minimal data bits, the weights can be transferred in the direction of the gradient if the erroe is in the correct direction. The error quantization is less intimately related to its own distribution than the activation value. Even when the error surpasses the maximum number of bits that can be represented, the network is still capable of learning; however, the gradient does not fall as rapidly as previously, therefore dynamic quantization of the error is unnecessary.

$$Backward: \quad E_{i8}^{(l)} = StQ\left(GEMM\left(StQ\left(SPMM\left(\widetilde{A}_{i8}^{(l)}, E_{i8}^{(l+1)}\right)\right), \left(W_{i8}^{(l)}\right)^T\right)\right) \tag{8}$$

## 3.6 QUANTIZATION OF GRADIENTS

Before entering the optimizer, the results of the most recent GEMM computation of radients are subjected to a dynamic quantization technique. As the gradient reduces during training, the error tensor eventually approaches the lower integer region, causing the estimated gradient after quantization to converge on 0. This phenomenon is especially evident in the backpropagation of the first layer, where the error of the first layer is always smaller in magnitude than the error of the second layer, and the transmission of the error in the first layer results in a very severe gradient disappearance as training continues. Consequently, we employ **DyQ** for the final GEMM of the calculated gradient to invert the offset of the error distribution throughout the training phase, as given in the Equation 9, and this alteration yields excellent results.

$$G_{i8}^{(l)} = DyQ\left(GEMM\left(\left(X_{i8}^{(l)}\right)^T, StQ\left(SPMM\left(\left(\widetilde{A}_{i8}^{(l)}\right)^T, E_{i8}^{(l+1)}\right)\right)\right)\right) \tag{9}$$

## 3.7 LOSS FUNCTION

The Softmax layer and cross-entropy criteria are extensively used in classification tasks, although the computation of $e^x$ cannot be utilized in instances of low-bitwidth linear mapping. We omit the Softmax layer and use the mean-square-error criteria instead (MSE), The back propagation of the loss function layer is shown in the Equation 10.

$$Backward: E_{i8}^{(last)} = \frac{\partial L}{X_{i8}^{(last)}} = X_{i8}^{(last)} - Y_{i8} \quad (Y_{i8} = Y_{onehot} \times 127) \tag{10}$$

# 4 EXPERIMENTS

Table 3: Comparison on the test accuracy/F1-micro, memory saving and training time.

| Node | 89K | | 232K | | 716K | | 1.5M | |
|---|---|---|---|---|---|---|---|---|
| **Edge** | 899K | | 114M | | 13M | | 264M | |
| **Dataset** | Flickr | | Reddit | | Yelp | | Amazon | |
| **Model** | Minibatch-GCN | | | | | | | |
| **Method** | FP32 | **GCINT** | FP32 | **GCINT** | FP32 | **GCINT** | FP32 | **GCINT** |
| **Acc(%)** | **50.10**±0.22 | 49.98±0.34 | **93.06**±0.08 | 92.91±0.12 | **38.14**±0.46 | 38.09±0.37 | 29.34±0.03 | **29.68**±0.07 |
| **Mem** | 1.52GB | **0.39GB** | 6.93GB | **1.79GB** | 2.75GB | **0.70GB** | 5.57GB | **1.48GB** |
| **Time/ep** | 1.99s | **0.08s** | 18.40s | **2.12s** | 27.90s | **1.93s** | 70.09s | **5.11s** |
| **Speedup** | 1× | **24.87×** | 1× | **8.68×** | 1× | **14.45×** | 1× | **13.70×** |
| **Node** | 2.7K | | 3.3K | | 19K | | 1.8K | |
| **Edge** | 10K | | 9K | | 88K | | 163K | |
| **Dataset** | Cora | | Citeseer | | Pubmed | | CS | |
| **Model** | Fullbatch-GCN | | | | | | | |
| **Method** | FP32 | **GCINT** | FP32 | **GCINT** | FP32 | **GCINT** | FP32 | **GCINT** |
| **Acc(%)** | 81.50±0.13 | **83.16**±0.37 | 71.26±0.14 | **71.68**±0.43 | **79.02**±0.17 | 78.86±0.39 | 91.10±0.25 | **91.50**±0.67 |
| **Mem** | 0.08GB | **0.02GB** | 0.83GB | **0.23GB** | 0.82GB | **0.21GB** | 1.36GB | **0.35GB** |
| **Time/ep** | 13.10ms | **0.32ms** | 12.20ms | **0.379ms** | 63.80ms | **3.69ms** | 104ms | **3.62ms** |
| **Speedup** | 1× | **40.94×** | 1× | **32.19×** | 1× | **17.29×** | 1× | **28.73×** |

Table 3 Comparison on the test accuracy/F1-micro, memory saving and training time for a single epoch on the 8 popular GNN benchmark datasets, more than 2,000 randomized trials have been undertaken. For the large graph, we employ the sampling-based mini-batch training method, where Flickr, Reddit, Yelp, and Amazon all use the hidden layer with 256 dimensions, 2 GCN layers and the GraphSage Hamilton et al. (2017) sampling method to fix 25 first-order neighbors and 10 second-order neighbors, and each dataset is randomly tested 50 times to determine the mean and standard deviation. Cora, Citeseer, Pubmed, and CS employ the full-batch training technique with a 16-dimensional hidden layer, 2 GCN layers and 500 random trials are conducted for each dataset to determine the mean and standard deviation. On the CUDA C++ side, the training time of GCINT is measured by invoking INT8 Tensor Core. Our method enables the GNN training process to take use of the GPU INT Tensor Core's considerable computational capacity, enabling relatively large-scale GNN models to be trained fast and affordably. Our code segment is in the Appendix.

# 5 CONCLUSION AND FUTURE WORK

In this work, we propose GCINT, a dynamic quantization training approach that adjusts the quantization range based on the sparsity of the graph. It can be employed in the GPU's INT Tensor Core to accelerate GCN training and reduce memory consumption without dequantization procedures. Meanwhile, GCINT is independent of the dataset and weight distribution. Experiments show that GCINT has at least 10× speedup and 4× memory consumption reduction with negligible accuracy loss compared with training in FP32 Cuda Core. Future work includes (1) implementing GCINT in existing framework; (2) combing GCINT with other memory-efficient training frameworks; (3) evaluating GCINT under other hardware accelerators; (4) evaluating GCINT under other neural networks.

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

# A    APPENDIX

```python
class GCINT_L0_GEMM(torch.autograd.Function):
    @staticmethod
    def forward(ctx, input, weight, nbit_W, nbit_A, nbit_G, nbit_E, Int,nbit_aa,nbit_ww,q_kw):
        Weight = Q_w(weight, nbit_W)
        output = torch.mm(input, Weight)
        output = DyQ(output)
        ctx.save_for_backward(Weight, input, nbit_G[0], nbit_E[0], Int[0],nbit_aa[0],nbit_ww[0])
        return output
    @staticmethod
    def backward(ctx, grad_output):
        Weight , input, nbit_G, nbit_E ,Int, nbitA,nbitw= ctx.saved_tensors
        nbitA = int(nbitA[0])
        nbit_W = int(nbitw[0])
        IntPoint = int(Int[0])
        nbit_G = int(nbit_G[0])
        nbit_E = int(nbit_E[0])
        input_T = torch.transpose(input,0, 1)
        Weight_T = torch.transpose(Weight,0,1)
        overall_grad = torch.mm(grad_output,Weight_T)
        Diff_W = torch.mm(input_T, grad_output)   #0.35s
        if IntPoint:
            Diff_W = DyQ(Diff_W)
            overall_grad = StQ(overall_grad, nbit_E, nbit_W)
        return overall_grad, Diff_W, None, None, None, None,None,None,None,None

class GCINT_L0_AGGREGATE(torch.autograd.Function):
    @staticmethod
    def forward(ctx, input_a, input_b, nbit_A,nbit_E, Int, nbit_aa,nbit_G):
        IntPoint = int(Int[0])
        output = torch.mm(input_a, input_b)
        if IntPoint:
            output = DyQ(output)
        ctx.save_for_backward(input_a, input_b, nbit_E[0],Int[0],nbit_aa[0],nbit_G[0])
        return output
    @staticmethod
    def backward(ctx, grad_output):
        input_a,input_b, nbit_E,Int ,nbit_aa,nbit_G= ctx.saved_tensors
        nBit_A = int(nbit_aa[0])
        IntPoint = int(Int[0])
        nbit_E = int(nbit_E[0])
        nbit_G = int(nbit_G[0])
        input_T = torch.transpose(input_a, 0, 1)
        overall_grad = torch.mm(input_T, grad_output)
        if IntPoint:
            overall_grad = StQ(overall_grad, nbit_E, nBit_A)
        return None, overall_grad, None, None, None,None, None,None
```

```python
class GCINT_L1_GEMM(torch.autograd.Function):
    @staticmethod
    def forward(ctx, input, weight, nbit_W, nbit_A, nbit_G, nbit_E, Int,nbit_aa,nbit_ww,q_kw):
        Weight = Q_w(weight, nbit_W)
        output = torch.mm(input, Weight)
        output = DyQ(output)
        ctx.save_for_backward(Weight, input, nbit_G[0], nbit_E[0], Int[0],nbit_aa[0],nbit_ww[0])
        return output
    @staticmethod
    def backward(ctx, grad_output):
        Weight , input, nbit_G, nbit_E ,Int, nbitA,nbitw= ctx.saved_tensors
        nbitA = int(nbitA[0])
        nbit_W = int(nbitw[0])
        IntPoint = int(Int[0])
        nbit_G = int(nbit_G[0])
        nbit_E = int(nbit_E[0])
        input_T = torch.transpose(input,0, 1)
        Weight_T = torch.transpose(Weight,0,1)
        overall_grad = torch.mm(grad_output,Weight_T)
        Diff_W = torch.mm(input_T, grad_output)   #0.35s
        if IntPoint:
            Diff_W = DyQ(Diff_W)
            overall_grad = StQ(overall_grad, nbit_E, nbit_W)
        return overall_grad, Diff_W, None, None, None, None,None,None,None,None

class GCINT_L1_AGGREGATE(torch.autograd.Function):
    @staticmethod
    def forward(ctx, input_a, input_b, nbit_A,nbit_E, Int, nbit_aa,nbit_G):
        IntPoint = int(Int[0])
        output = torch.mm(input_a, input_b)
        if IntPoint:
            output = DyQ(output)
        ctx.save_for_backward(input_a, input_b, nbit_E[0],Int[0],nbit_aa[0],nbit_G[0])
        return output
    @staticmethod
    def backward(ctx, grad_output):
        input_a,input_b, nbit_E,Int ,nbit_aa,nbit_G= ctx.saved_tensors
        nBit_A = int(nbit_aa[0])
        IntPoint = int(Int[0])
        nbit_E = int(nbit_E[0])
        nbit_G = int(nbit_G[0])
        input_T = torch.transpose(input_a, 0, 1)
        overall_grad = torch.mm(input_T, grad_output)
        if IntPoint:
            overall_grad = StQ(overall_grad, nbit_E, nBit_A)
        return None, overall_grad, None, None, None,None, None,None

class MyDropout(torch.autograd.Function):
    @staticmethod
    def forward(ctx, input, rate):
        Rate_tensor = (rate*torch.ones(input.shape[0], input.shape[1])).cuda()
        random_tensor = torch.rand(input.shape[0], input.shape[1]).cuda()
        Mask_tensor = torch.ge(random_tensor, Rate_tensor).cuda()
        output = torch.mul(Mask_tensor, input)
        ctx.save_for_backward(Mask_tensor)
        return output
    @staticmethod
    def backward(ctx, grad_output):
        result = ctx.saved_tensors
        grad = result[0]
        overall_grad = torch.mul(grad_output, grad)
        return overall_grad, None
```

```python
class MyReLu(torch.autograd.Function):
    @staticmethod
    def forward(ctx, input, nbit_E,Int):
        output = F.relu(input)
        ctx.save_for_backward(output, nbit_E[0],Int[0])
        return output
    @staticmethod
    def backward(ctx, grad_output):
        output ,nbit_E, Int= ctx.saved_tensors
        IntPoint = int(Int[0])
        nbit_E = int(nbit_E[0])
        data = output
        grad = torch.zeros(data.shape[0],data.shape[1]).cuda()
        grad = torch.ne(grad,data)     #not equal
        overall_grad = torch.mul(grad_output,grad)
        if IntPoint:
            overall_grad = overall_grad
        return overall_grad , None,None

def Q_w(w, nbit):
    w = torch.round((w*127) / (torch.max(torch.abs(w))))
    return w

def StQ(w, nbitA,nbitW):
    A = ((2**(nbitA+nbitW)))
    w = torch.clamp(w,-A,A)
    w = torch.round(w/(2**(nbitW)))
    return w

def DyQ(input):
    kw = check(input)
    A = int((2 ** (kw)))
    w = torch.clamp(input, -A, A)
    w = torch.round(w / (2 ** (kw - 7)))
    return w

def check(input):
    reg = []
    reg_nub = []
    input = abs(input)
    for i in range(1,32):
        temp = torch.ones(input.shape[0],input.shape[1])*(2**i)
        temp = temp.to('cuda')
        mid = torch.ge(temp,input)
        reg.append(int(torch.sum(mid)))
        if i == 1:
            reg_nub.append(reg[i-1])
        else:
            reg_nub.append(reg[i-1] - reg[i - 2])
    local_pum = []
    for k in range(7,32):
        local_pum.append(sum(reg_nub[k-7:k]))
    k = np.argmax(local_pum) + 7
    return k
```

