# OpenReview forum: "GCINT: Dynamic Quantization Algorithm for Training Graph Convolution Neural Networks Using Only Integers"
_ICLR.cc/2023/Conference — Submitted to ICLR 2023_

### Official Review · Reviewer_Po7t · 2022-10-24

**Confidence:** 4
**Correctness:** 1
**Technical Novelty And Significance:** 2
**Empirical Novelty And Significance:** 1
**Recommendation:** 3

**Clarity, Quality, Novelty And Reproducibility:**

* Clarity:

  * I found the paper hard to follow, as explanations seem convoluted and structure needs improvements. Some examples:

    * On page 2, " a shaping model can be obtained directly". What do the authors understand by a "shaping" model?
    * On page 2, what do the authors understand by "quantification methodology"?
    * The authors refer in several places to "dequantization" from FP to INT (e.g. on page 3,
    "the output [...] must be dequantized from the FP32 to INT16 format"). Shouldn't it be quantization that is needed
    here?
    * On page 4, "$x_{fn}$ represents the signed <i>bitn</i> to which $x$ is quantized". I am confused as to what a
    "bitn" is.
    * I guess $x_{in}$ and $x_{bn}$ are both the integer representation of $x$, just that the first is viewed as an
    integer in base 10, and the other is viewed in binary. It is confusing to use different mathematical notations
    for the exact same mathematical object (the same integer).
    * In Figure 3, they authors plot a histogram of a tensor multiplication output, but they do not state how was this
    histogram obtained. Were the tensors sampled from a certain distribution? Or is this obtained from a training/test
    sample?
    * The function $f$ appearing in equation (2) is never introducted (the only place it appears is on the vertical axis
    of figure 3).
    * The authors mention that "We utilize Kullback-Leibler divergence (Joyce, 2011) to prove that when g(k) in
    Equation 2 gets the maximum, we may derive a 7-bit(k, k + 6) expression for this data set with reduced Kullback-
    Leibler divergence and more Shannon Entropy (Bromiley et al., 2004)", but no proof is provided, nor is it clear how
    KL divergence/Shannon entropy is related to maximizing g(k).
    * In Algorithm 1:
      * line 4, who is $j$? This variable is never iterated over nor assigned anywhere.
      * line 5, AdderTree has a height argument, but what output is it invoked over?
      * LoadAtoeachThreadGroup and AdderTree are never defined.
      * The vector $C$ is never initialized.

* Novelty:

  * From a technical point of view, the framework is very similar to the NITI method, see
  <i>M. Wang, S. Rasoulinezhad, P. Leong and H. So. NITI: Training Integer Neural Networks Using
  Integer-Only Arithmetic,  IEEE Transactions on Parallel and Distributed Systems, 2020, DOI:10.1109/TPDS.2022.3149787
  </i>. While not applied to GNNs, their training pipeline works on full 8-bit integer arithmetic, and their dynamic
  shift and shift-and-round criterion is almost identical, with the only difference that they select the shift
  count to minimize overflow alone, while here the authors also take into account underflow. From a technical
  standpoint, this work is just an incremental step over their method.
  * There is no empirical novelty, as this work simply compares the integer-only training pipeline against the FP32
  baseline on standard GNN datasets.

* Quality:

  * Relevant baselines are not compared against (see baselines mentioned in section 2 and the NITI reference above)
  and no analysis of results is provided.
  * No theoretical justification for the criterion is provided (minimizing KL divergence is mentioned, but no proof is
  given).
  * Overall, the scientific quality of this work, in its present form, is below acceptance standards for ICLR.

* Reproducibility:

  * The authors provide some information on their experimental procedure in the caption of Table 3.
  * Some code snippets are provided in the manuscript, but no intention to release publicly available code is
  mentioned. They do mention "implementing in existing framework" as a future intention, but it is not clear what
  framework they are referring to or whether this means the code will be public.

**Strength And Weaknesses:**

* Strengths:

  * Due to the scalability issues and online-training demands for, integer-only training and inference is likely to have
  high practical impact.

* Weaknesses:

  * Technical contribution is incremental (see notes below on novelty).
  * Clarity could be improved a lot (see remarks below on clarity).
  * No comparison is provided against any competing method. The only baseline is the floating point pipeline.


**Summary Of The Paper:**

This work proposes a method to accelerate both training and inference in Graph Neural Networks by quantizing both the
weights and all the intermediate results (features, errors, gradients) to 8 bits. The proposed training and inference
pipeline can be executed on integer-only hardware.

To deal with the high dynamic range arising in GNNs due to various node degrees and feature types, this work proposes
to use a dynamic quantization technique to map 32-bit intermediate results from multiply-and-accumulate steps back to
8-bits shifting right with a dynamically-determined amount and clipping to 8 bits. The shift count is determined such
that the number of outliers resulting in overflow or underflow is minimized.

Some experiments are carried out on 8 GNN benchmarks, matching the accuracy of floating point counterparts. No
comparisons against state-of-the-art GNN quantization methods are provided.

**Summary Of The Review:**

This seems to be unfinished work with incremental technical novelty. The manuscript needs a major rewrite to
meet the presentation quality standards for publication (see notes on clarity above). At lease one relevant baseline is
missing (see e.g. NITI above), and no comparison against any competing work is provided. Overall,
in its current form, this work does not meet acceptance standards for ICLR.

---

> ### Author Response · Authors · 2022-11-18
> **Response**
>
> Dear reviewers:
> Thank you very much for reading our paper and for your insightful revision suggestions. The paper contains a big number of wrongly written symbols and comments; we will make modifications in response to your feedback. We regret that we did not do research on the NITI piece, but there are significant distinctions between NITI's approach and our own. In our prior tests, we also investigated employing a scaling factor similar to that of NITI, i.e., utilizing the largest value as the scaling factor. However, it did not perform well on GNN, so we presented an enhanced version of the technique (DyQ). We will upload the algorithm's source code to github and modify it in response to your feedback. Your comments are very thoughtful, which is excellent; we appreciate your feedback on our work.
> Yours Sincerely

---

### Official Review · Reviewer_5yWh · 2022-11-04

**Confidence:** 3
**Correctness:** 3
**Technical Novelty And Significance:** 3
**Empirical Novelty And Significance:** 3
**Recommendation:** 3

**Clarity, Quality, Novelty And Reproducibility:**

The writing of this paper should be improved for general readers.

Though the quality of writing is not satisfactory, I feel this paper contributes some technical novelty.

The code snippets are provided for reproducibility.


**Strength And Weaknesses:**

**Strength**
- The quantization of GNNs is an important problem, and the paper provides a comprehensive comparison of different quantization methods.
- Different quantization operators are proposed to handle real and pseudo overflows, and are plugged into the GNN workflow.
- The empirical results show improvement in efficiency.

**Weaknesses**
- Though I believe this paper has some nontrivial technical contributions, the writing of this paper needs to be greatly improved to make it accessible for the community. I have tried hard and spent a significant amount of time, but can still get a rough idea without understanding the details of why pseudo flow is a key bottleneck and how it is handled by DyQ. For example, Section 3.1 is hard to follow for a general reader like me: what exactly the pseudo overflow problem is (e.g. when and why $c$ cannot be represented by $2^n-1$), what is the correlation and difference between real and pseudo overflow. Figure 2 is almost impossible to understand for general readers too.
- It is not clearly justified why dequantization is not needed in this method.
- The claim that this method can accelerate inference is not supported by experiments.
- This method is not compared with existing works, for example, the memory reduction compared with EXACT.


**Summary Of The Paper:**

This paper proposes four factors that restrict the applicability of traditional quantization methods in GNNs. Based on that, a specialized and efficient quantization framework is proposed to improve forward, backward, optimizer, and loss functions with the dynamic quantization (DyQ), static quantization (StQ), and weight quantization (Qw ) operators. Experiments show the reduced memory and time complexity with comparable accuracy.


**Summary Of The Review:**

This paper identifies some nontrivial challenges in quantization of GNNs. However the writing of this paper is hard to follow for a larger audience. I believe it is a work with good potential if it can be better presented.

---

> ### Author Response · Authors · 2022-11-18
> **Response**
>
> Dear reviewers:
> We appreciate you reading our paper and providing us with your helpful feedback. It is true that several of the article's sections are not particularly well written, thus we will revise the paper and provide comparative trials using others approaches. I want to thank you again for your feedback.
> Yours Sincerely

---

### Official Review · Reviewer_o9Df · 2022-11-10

**Confidence:** 3
**Correctness:** 2
**Technical Novelty And Significance:** 2
**Empirical Novelty And Significance:** 2
**Recommendation:** 3

**Clarity, Quality, Novelty And Reproducibility:**

The writing is reasonably clear, but some claims which was to justify the proposed method were not demonstrated.


**Strength And Weaknesses:**

Pro: The paper claims to improve training speed by using dynamic quantization.

Con: The solution is complex, which limits its applicability. And technical novelty is rather limited.


**Summary Of The Paper:**

The work proposes quantization methods for low bit training of GNNs.

**Summary Of The Review:**

The paper is not ready for publication. Additional works are needed to justify the claims and to simplify the algorithms.

---

> ### Author Response · Authors · 2022-11-18
> **Response**
>
> Dear reviewers:
> We are very grateful to your comments for the manuscript. According with your advice, we tried our best to amend the relevant part and made some changes in the manuscript. In later versions, the algorithm's complexity will be as minimally complicated as possible. I want to thank you again for your feedback.
> Yours Sincerely

---

### Decision · Program_Chairs · 2023-01-20

**Decision:**

Reject

**Justification For Why Not Higher Score:**

All reviewers rejected

**Justification For Why Not Lower Score:**

N/A

**Metareview: Summary, Strengths And Weaknesses:**

This work proposes a method to accelerate GNNs during training and inference by using dynamic quantization. The method experimentally is effective, but no comparison is given with state-of-the-art competing methods, and the technical novelty is limited with respect to existing approaches to network quantization, e.g. the NITI method of Wang et al. in IEEE Transactions on Parallel and Distributed Systems, 2020.